# Model of Oxygen Conditions within Aquaculture Sea Cages

**DOI:** 10.3390/biology12111408

**Published:** 2023-11-08

**Authors:** Heiðrikur Bergsson, Morten Bo Søndergaard Svendsen, John Fleng Steffensen

**Affiliations:** 1Marine Biological Section, Department of Biology, University of Copenhagen, DK-3000 Elsinore, Denmark; mosv@di.ku.dk (M.B.S.S.); jfsteffensen@bio.ku.dk (J.F.S.); 2Hiddenfjord, Við Ánna 1, FO-512 Norðragøta, Faroe Islands; 3Copenhagen Academy for Medical Education and Simulation, Rigshospitalet, Capital Region of Denmark, DK-2100 Copenhagen, Denmark; 4Department of Computer Science, University of Copenhagen, DK-2100 Copenhagen, Denmark

**Keywords:** aquaculture, model, swimming energetics, hypoxia, oxygen availability

## Abstract

**Simple Summary:**

This study establishes how trout’s swimming energy changes under various oxygen levels and temperatures, leading to the development of a model that predicts oxygen conditions inside the cages. A case study using the model reveals that a 10 °C rise in temperature causes oxygen concentration to drop threefold, necessitating a 3.7-fold increase in water flow to maintain optimal oxygen levels.

**Abstract:**

To ensure optimal feed intake, growth, and general fish health in aquaculture sea cages, interactions between drivers that affect oxygen conditions need to be understood. The main drivers are oxygen consumption and water exchange, caused by flow through the cage. Swimming energetics in rainbow trout (*Oncorhynchus mykiss*) in normoxia and hypoxia at 10, 15, and 20 °C were determined. Using the determinations, a conceptual model of oxygen conditions within sea cages was created. By applying the model to a case study, results show that with a temperature increase of 10 °C, oxygen concentration will decrease three times faster. To maintain optimal oxygen concentration within the cage, the flow velocity must be increased by a factor of 3.7. The model is highly relevant for current farms since the model predictions can explain why and when suboptimal conditions occur within the cages. Using the same method, the model can be used to estimate the suitability of potential new aquaculture sites.

## 1. Introduction

Oxygen availability is a critical parameter in ensuring feed intake, optimal growth, and health in cultured fish [1,2,3,4]. The primary drivers for oxygen availability within aquaculture sea cages can be divided into slow- and fast-changing parameters, i.e., variations occurring seasonally to daily or hourly to min^−1^. In the present study, the slow-changing parameters are temperature, salinity, and atmospheric pressure, directly affecting oxygen solubility [5]. The fast-changing parameters are the current velocity and oxygen consumption ($\dot{M}O_{2}$) of the fish. The balance between these two fast-changing parameters is essential in retaining suitable oxygen conditions in the cages since $\dot{M}O_{2}$ depletes oxygen while the current velocity replenishes it. 

The $\dot{M}O_{2}$ ranges from the standard metabolic rate (SMR) to the maximum metabolic rate (MMR), where SMR is the oxygen consumption needed to sustain life, i.e., a resting fish with no post-prandial effect and MMR is the maximum $\dot{M}O_{2}$ [6,7,8]. Measurements of MMR are achieved during the maximum swimming velocity, when the fish utilizes bursting, which results in anaerobic metabolism and therefore is not sustainable for prolonged periods. The $\dot{M}O_{2}$ for maximum sustained swimming speed is referred to as active metabolic rate (AMR) and utilizes aerobic metabolism [6,9]. The difference between SMR and AMR is the aerobic metabolic scope (AMS) and is defined as the respiratory availability for all activities excluding SMR.

Fish have the option to swim at different swimming speeds, ranging from hovering to the maximum swimming speed, i.e., the critical swimming speed (U_crit_) [10]. In a confined area, such as a sea cage, this range narrows with increased current velocity and during high water flow, the only option for the fish is to swim against the current. During low water flow, the fish can regulate swimming at an intermediate speed which can be defined as the speed which the fish chooses to swim at and is denoted as the preferred swimming speed (U_pref_). There are likely multiple interrelated reasons for choosing this speed and a study [11] found a possible answer in U_pref_ being equal to the optimal swimming speed (U_opt_) in brook charr (*Salvelinus fontinalis*) at 15 °C. The U_opt_ is determined from the cost of transport (COT) [12] and is defined as the speed at which the cost of swimming a given distance is the lowest, hence giving an energetically saving explanation to the chosen speed. Although plausible, U_opt_ has previously been shown to increase with temperature [13,14]. For U_pref_ to increase in the same manner as U_opt_, the fish would need to opt for expending an extra amount of energy to attain the minimal cost of transport, i.e., U_opt_. Therefore, this seems improbable and more likely U_pref_ remains somewhat unchanged across the temperature range of salmonids.

A possible explanation to why U_opt_ was found to equal U_pref_ in a previous study [11] might be found in the optimum temperature (T_opt_) [15] for brook charr, which is the same as the experimental temperature (15 °C) [16]. Using this assumption, the model in the present study will use U_opt_ at close to T_opt_ to determine U_pref_.

For the model to reflect swimming activity within an active aquaculture site, normal operations, such as feeding and maintenance need to be considered, since they contributed to increased activity/swimming speed and fish distribution within the cages [17,18].

During feeding events, a substantial increase in activity can be expected and as $\dot{M}O_{2}$ increases exponentially with increasing swimming speed [10], the oxygen demand will increase correspondingly. Furthermore, if water exchange is insufficient, this combination will cause decreases in oxygen levels, especially in the feed region, i.e., the top layer of the water column. Although significant for the model outcome, due to simplicity reasons, these local events are not included in the model but should be considered in the model outcome. 

As with increased swimming speed, $\dot{M}O_{2}$ also increases with higher temperature. The increase can be quantified using the factorial increase in $\dot{M}O_{2}$ with a temperature increase of 10 °C (Q10). In fishes, Q10 ranges from 1.62 to 2.64 [8,19,20,21,22].

Increases in $\dot{M}O_{2}$ are not solely caused by external factors but also by internal factors such as digestion. Digestion or postprandial effects will increase the $\dot{M}O_{2}$ for several hours after feeding depending on temperature, portion size, and species [23,24]. The increase is referred to as specific dynamic action (SDA). The magnitude of SDA after feeding changes over time, with an initial increase in $\dot{M}O_{2}$, reaching a peak and thereafter decreasing back to SMR [23,25,26]. In most aquaculture facilities, fish are fed at regular time intervals at least once a day. A study [27] found the duration of SDA in rainbow trout fed 1% of body mass at 10–11.5 °C to be around 50 h. The duration between the feeding intervals is usually less than the digestion time for rainbow trout and therefore, $\dot{M}O_{2}$ will continuously be elevated, essentially reaching an equilibrium over time.

In the present study, we suggest that feeding can be optimized by modeling and understanding oxygen availability within sea cages and using the outcome of the model to plan feeding. If oxygen concentration decreases below a threshold, the feed is sub-optimally utilized, and the energetic value of feed consumed is reduced [28,29] Furthermore, prolonged decrease in oxygen concentration within the cage will cause loss of appetite [30,31].

A conceptual model was made to understand how fish and the environment affect oxygen availability within sea cages. With this understanding excess feed, which is not utilized by the fish, can be reduced, and thereby reducing the effects on surrounding environment, optimizing feed conversion ratio and finally increase fish health within the cage. Furthermore, in Danish aquaculture, production in sea cages is limited by the discharge from aquaculture farms [32]; hence if the discharge is reduced, production can be increased, allowing the farmers to increase their production, with their current farming permits.

## 2. Materials and Methods

### 2.1. Experimental Fish

Rainbow trout (*Oncorhynchus mykiss* (Walbaum, 1792)) were chosen as the model species since it is the only species farmed in Danish sea cage aquaculture (Danish Fisheries Agency statistics 2021). Fish were obtained from a land-based aquaculture farm (Sten Kjær ApS, Brørup, Denmark). Due to the relatively large fish size (1510.8 ± 479.7 g) and limited space in the laboratory, fish were obtained on three separate occasions, approximately 30 fish in each batch. Unfortunately, the fish farm was not able to deliver the same size fish on the three separate occasions.

Fish were kept in two flow-through holding tanks, with 20 fish in a 1300 L circular tank and ten fish in a square 700 L tank, both connected to the laboratory system water. The system provided fully aerated (95% > air saturation dissolved oxygen (air sat. DO)), recirculated, filtered seawater kept at 10 °C, and salinity of Approx.. 30 ppt. The light:dark regime was 12:12 h. Pressurized air was bubbled through the water column to ensure that CO_2_ did not exceed the recommended levels [33].

### 2.2. Acclimation

All experiments were conducted in seawater with a salinity of 30 ppt. Since fish had been reared in freshwater, acclimation to the new salinity gradient was allowed before experiments could be performed. The acclimation was set to a minimum of 4 weeks since the state of smoltification was unknown. Smoltified salmonids can effectively acclimate to new salinities within days [34,35]. Initially, fish would not eat, and therefore feeding was postponed for two weeks. After that, fish were fed 8 mm commercial trout pellets (BioMar A/S, Brande, Denmark) twice a week. When the initial acclimation period was over, the fish were tagged using sterile 8 mm PIT tags (ADEQID microchip tags, Eickemeyer Aps, Vojens, Denmark). PIT tags do not affect swimming performance in rainbow trout [36]. After tagging, fish were size sorted for the experiments and returned to the two holding tanks.

Since the experiments required fish to be acclimated to different temperatures, after salinity acclimation, batch two and three were acclimated to 15 and 20 °C for a minimum of four weeks [37], respectively. During this acclimation period, all fish were monitored for general health, especially for infections at the PIT tag puncture wound. For consistency, the 10 °C batch was also kept for four weeks before experiments could commence. The protocol for temperature acclimation was to increase the temperature by 1 °C·day^−1^ until the target temperature was reached.

Three 500 W titanium heaters (TH-500, Aqua Medic Direct, Leicestershire, United Kingdom) were mounted vertically in the holding tank and controlled by a programmable thermoregulator (PR5714, PR Electronics, Rønde, Denmark) programmed to heat one °C above the benchmark temperature (15 and 20 °C). To ensure cooling and water exchange, a second thermoregulator (PR5714, PR Electronics, Rønde, Denmark), programmed to turn on when the benchmark temperature was reached, controlled two pumps (56 L/min, EHEIM GmbH, Deizisau, Germany), placed in an adjacent tank and connected to the setup. The adjoining tank did not contain fish and was continuously supplied with system water. Using this method, the temperature was maintained at 15.0 ± 0.2 and 20.0 ± 0.2 °C. Although the method is sufficient at maintaining the temperature, water exchange is decreased, and waste product build-up can become an issue. During the present study, this was not deemed an issue due to the small number of fish in the tank and reduced diet.

### 2.3. Study Design

In the present study, swimming energetics in normoxia and hypoxia were determined using ten replicates at three different temperatures. Carryover effects on SMR between the experiments were tested using a crossover protocol, i.e., five fish exposed to normoxia than hypoxia, and five fish in a reversed protocol for each temperature.

To verify if SDA signal reaches an equilibrium over time during regular feeding, experiments were conducted on 12 fish simultaneously. To reduce external disturbance during experiments, all setups were covered by dark fabric and remotely monitored via video cameras. Finally, acquired data were used to create a model of oxygen availability within an aquaculture sea cage.

#### 2.3.1. Swimming Energetics—Normoxia

Swimming energetics and critical swimming speed were determined using a swim tunnel respirometer (90 L Steffensen type) with a swim section of 20 × 20 × 70 cm (h, w, l). Dissolved oxygen was measured using a fiber-optic oxygen meter (Fibox 3, Precision Sensing GmbH, Regensburg, Germany) without temperature compensation. The oxygen meter was recalibrated between each temperature change. Oxygen consumption was calculated using data acquisition software (AutoResp, Loligo Systems, Tjele, Denmark) which controlled the flush, wait, and measuring periods (600 s, 180 s, and 720 s), resulting in one measurement every 25 min. Flow in the respirometer was generated using a propeller connected to an electric motor (AC-motor, DRS71, SEW Eurodrive, Bruchsal, Germany), controlled by a motor controller (Movitrac MCLTE, B0004-101-1-20, SEW Eurodrive, Bruchsal, Germany). Calibration of flow velocity used to calculate swimming speed (BL·s^−1^) was measured using a flow probe (Vane Wheel, Höentzsch, Weiblingen, Germany). Using the measured flow and the corresponding mV signal, a regression line was fitted through the points resulting in the flow calibration equation.

To ensure minimum temperature fluctuations in the swim tunnel, the same setup used in the holding tank was applied. Experimental temperatures were 10, 15, and 20 °C.

Postprandial effects on oxygen consumption were reduced by fasting the fish for a minimum of 24 h before introduction to the respirometer [26]. During the first 24 h in the respirometer, the fish were swimming at 0.4 BL·s^−1^. Initially, $\dot{M}O_{2}$ measurements showed a steep decline, followed by a plateau. The swimming protocol had one determination per increment with increments of 0.3 BL·s^−1^ until the fish fatigued.

The critical swimming speed (U_crit_) was determined as the point at which a fish would continuously rest on the swim section’s back-grid after three consecutive attempts at encouraging swimming by shortly decreasing flow velocity. When U_crit_ was reached, the experiment was terminated. 

Bacterial respiration and accumulation of waste products were reduced by emptying and cleaning the setup between each experiment. To check for bacterial background respiration, oxygen measurements in the swim tunnel without fish were conducted on multiple occasions and were always negligible.

#### 2.3.2. Swimming Energetics—Hypoxia

The swimming energetics in hypoxia were conducted in the same setup as the swimming energetics in normoxia. Hypoxia was achieved by utilizing the method of self-inducing hypoxia allowing the fish to decrease the water’s ambient oxygen tension [38]. As opposed to bubbling with nitrogen, self-induced hypoxia to reduce oxygen tension will cause waste products to accumulate in the setup. Waste product build-up will eventually affect the performance of the swimming fish and therefore be a more realistic scenario of what happens in a sea-cage during periods with slow-moving water (limited exchange) [39,40].

To determine the minimum oxygen tension required for swimming at different swimming speeds, experiments were completed at 0.4 and 1.0 BL·s^−1^. The critical oxygen tensions (P_crit_) were determined as pO_2_ when the fish stopped swimming and rested on the swim section’s back grid. The hypoxia experiments were initiated by changing the flush and wait periods to one second, after an initial 24 h acclimation period with fish swimming at 0.4 BL·s^−1^. After P_crit_ was determined for 0.4 BL·s^−1^, the flush and wait periods were changed back to pre-experimental settings and left for 24 h. The second part of the hypoxia trial was initiated by increasing the swimming velocity to 1.0 BL·s^−1^ with 0.3 BL·s^−1^ increments. As soon as the fish displayed normal swimming behavior at 1.0 BL·s^−1^, the experiment was initiated using the same protocol as in the first experiment.

#### 2.3.3. SDA Experiments

To quantifying the SDA signal using $\dot{M}O_{2}$ and verifying the hypothesis that if feed is administered with regular intervals, the SDA signal will reach an equilibrium, a respirometer consisting of a circular tank with a water volume of 1342 L was used. The water temperature was 10 ± 0.2 °C and salinity 30 ppt. 

Open-tank respirometry is affected by gas-exchange at the tank’s surface, and the exchange should therefore be quantified. By turning off the water supply and bubbling nitrogen through the water column, the oxygen tension was reduced to approximately 50% air sat. DO. Using an optical oxygen meter (FireStingO2, PyroScience GmbH, Aachen, Germany), the oxygen content was measured and logged until equilibrium between water and air was reached. No fish were in the tank during this experiment.

Respirometry measurements were calculated using the freeware AquaResp (www.aquaresp.com, DOI: 10.5281/zenodo.2584015, last accessed: 6 March 2021), which also controlled Flush, Wait, and Measuring durations (F: 900 s, W: 120 s, M: 1380 s) using a USB-Switch C mechanical relay (Cleware GmbH, Hollingstedt, Germany). Two pumps (56 L/min, EHEIM GmbH, Deizisau, Germany) connected to the respirometer tank were placed in an adjacent tank. The adjacent tank was continually supplied with system water and did not contain any fish. During the flushing period, the relay also opened a solenoid valve that controlled airflow to air stones ensuring degassing of CO_2_ in the respirometer.

The combined mass of all fish was 23.7 kg, giving a fish density of 17.7 kg/m^3^. In aquaculture cages, fish size variation is common, and 12 fish ranging from 781 g to 3074 g with an average mass of 1976.4 ± 667.9 g (average ± SD) were chosen for the experiment.

To ensure that the expected increase in $\dot{M}O_{2}$ caused by SDA could be measured in the open-tank respirometer, 3 replicates were performed using one feeding of 2% of total body mass (TBM). Once completed and verified, the experiment to quantify $\dot{M}O_{2}$ caused by SDA at equilibrium with continual daily feeding (1% TBM) could be performed. For consistency an automatic fish feeder (XClear, Son en Breugel, The Netherlands) was used. The feeder was programmed to dispense the predefined ration over an hour from midday. The extended feeding duration prevented pellets from accumulating on the bottom and ensured that all pellets were consumed. During the first 24 h in all experiments, no feed was administered.

### 2.4. Theory/Calculations

Applying the measured parameters from the present study into a model required data analysis. The data were analyzed using Python 3.8.5 with data analysis packages. The model was likewise created using Python 3.8.5.

#### 2.4.1. Swimming Energetics—Normoxia

Swimming energetics (*x*: Swimming speed; *y*: $\dot{M}O_{2}$) during normoxia were fitted to the following exponential function.
(1)$$y=a\;\cdot e^{\left(b\;\cdot x\right)},$$

From the exponential function, cost of transport (COT) is calculated by dividing $\dot{M}O_{2}$ with the swimming speed. The lowest COT, i.e., the speed at which the cost of swimming one body length is the lowest is referred to as the optimal swimming speed (U_opt_). U_opt_ is determined as the tangent to the fitted regression (Equation (1)) and is expressed using the following equation [40].
(2)$$U_{opt}=b^{-1},$$
where *b* is the resulting exponent from the fitted swimming equation (Equation (1)).

The critical swimming speed (*U_crit_*) was calculated using the following equation [10].
(3)$$U_{crit}=u_{i}+\left(\frac{t_{i}\;\cdot u_{ii}}{t_{ii}}\right),$$
where *u_i_* is the swimming speed maintained throughout a completed increment (BL·s^−1^), *t_i_* is the duration spent swimming at the incomplete increment (min), *t_ii_* is the duration of the whole incomplete increment (min), and *u_ii_* is the velocity increase between increments (BL·s^−1^).

To compare $\dot{M}O_{2}$ at different temperatures in the model, the considerable variation in fish mass had to be mass adjusted. The SMR and AMR were body mass adjusted using the method described in [41]. The chosen target body mass was 1.5 kg, close to the average for all fish (1452.7 ± 490.1 g).
(4)$$\dot{M}O_{2\;\left(1.5kg\right)}=\dot{M}O_{2}\cdot (\frac{M}{1.5}{)}^{\left(1-A\right)},$$

$\dot{M}$O_2(1.5kg)_ is the adjusted oxygen consumption rate when fish mass is 1.5 kg, $\dot{M}O_{2}$ is the measured oxygen consumption (mg O_2_ kg^−1^ h^−1^), M is the body mass, and A is the mass exponent. The mass exponent chosen in the present study was 0.89, which was found suitable for rainbow trout [42]. Finally, the factorial aerobic metabolic scope (FAMS) is calculated by dividing AMR with SMR.

#### 2.4.2. Swimming Energetics—Hypoxia

Data from the hypoxia experiments (swimming at 0.4 and 1.0 BL·s^−1^ and AMR from normoxia experiments) were fitted to the Hill equation [43]. The Hill equation is a sigmoid function and is expressed as follows:(5)$$AMS_{Hypoxia}=(\frac{AMR}{1+(\frac{b}{x}{)}^{c}}),$$
where AMS*_Hypoxia_* is the aerobic metabolic scope during hypoxia, *AMR* is the Active Metabolic Rate, *b* is the P50, *c* is the Hill coefficient, and *x* is the partial pressure of oxygen (pO_2_). Applying the $\dot{M}O_{2}$ during Upref, the Hill equation can estimate the minimum pO_2_ required to swim at Upref. Constants used in the Hill equation for 10, 15, and 20 °C were P50: 6.50, 6.04, and 6.24 and Hill coefficient: 3.50, 5.51, and 5.70, respectively.

#### 2.4.3. SDA Experiments

To quantify the increase in $\dot{M}O_{2}$ caused by the SDA at equilibrium, the baseline $\dot{M}O_{2}$ of all fish was acquired by measuring oxygen consumption over 24 h pre feeding. Since the fish were free to swim around during the experiment, the acquired baseline for $\dot{M}O_{2}$ is referred to as routine metabolic rate (RMR). Large variations in $\dot{M}O_{2}$ were measured during the daytime, but during nighttime, the measurements stabilized. Therefore, the RMR baseline in the experiments was determined as the median value of all night measurements. 

#### 2.4.4. The Model

The model consists of three main calculation series, which describe the oxygen conditions within an aquaculture cage during changes in biological and environmental factors. All calculation series are calculated for fed and unfed fish.

The oxygen solubility is calculated as described in [5]. The sizes of circular cages in aquaculture are usually referenced using the circumference of the cage. Therefore, the cage volume is calculated using circumference and depth as in the following equation:(6)$$Vol=\pi \;\cdot (\frac{c}{2\;\cdot \;\pi }{)}^{2}\;\cdot d,$$
where *Vol* is the volume, *C* is the circumference, and d is the depth of the cage. The equation is a combination of the equation for the volume of a cylinder and radius from the circumference. Using the cage volume, the fish density is calculated using the following equation:(7)$$\rho =\frac{Q\;\cdot m}{Vol},$$
where ρ is the fish density, *Q* is the total number of fish, m is the average mass of the fish, and Vol is the cage volume.

The first calculation series (CS1) derives the time for oxygen concentration to reach anoxia with fish at rest and swimming at U_pref_ and with no water flow through the cage. The series also calculates the duration to oxygen depletion when fish are fed, i.e., $\dot{M}O_{2}$ during stable SDA.

The average oxygen consumption for resting fish and fish swimming at U_pref_ when fed and unfed was subtracted from the ambient oxygen concentration over time. This can be expressed using the following equation:(8)$$y_{t}={\left[O_{2}\right]}_{cage}-\dot{M}O_{2}\cdot \;\rho \cdot t,$$
where *y_t_* is oxygen concentration at time t anywhere in the cage with no flow and [O_2_]_cage_ is the oxygen concentration in the cage, $\dot{M}O_{2}$ is the oxygen consumption, ρ is the fish density within the cage, and t is time. Since fish cannot sustain swimming at U_pref_ below the critical oxygen concentration, found using the Hill equation (Equation (4)), the model calculates a lower swimming speed and subsequently the $\dot{M}O_{2}$. The updated $\dot{M}O_{2}$ is the result of the Hill equation using the oxygen concentration in the cage.

The second calculation series (CS2) calculates the oxygen concentration distribution through the cage, given a preset current velocity and swimming speed in fed and unfed fish. This is achieved using the following equations:(9)$$y_{x}=[O_{2}{]}_{cage}-\left(\dot{M}O_{2\;(sec)}\cdot \;\rho \cdot v^{-1}\right)\cdot \;D,$$
where *y_x_* is the oxygen concentration at distance x from the upstream edge of the cage and [O_2_]*_cage_* is the oxygen concentration in the cage, $\dot{M}O_{2(\sec)}$ is the oxygen consumption in seconds, ρ is the fish density, v^−1^ is the time for the current to move one meter, and D is the diameter of the cage. As in the first series, a limit is added to the swimming velocity if oxygen concentration decreases below the minimum requirement for swimming at U_pref_.

The third calculation series (CS3) calculates the oxygen concentration based on two presets. The first preset is when current velocity and swimming speed are the same and the second when swimming speed is at U_pref_ until current velocity exceeds U_pref_. To compare $\dot{M}O_{2}$ at different swimming speeds with the corresponding current velocities, the units must be unified, i.e., convert swimming speed from BL·s^−1^ to cm·s^−1^ using the fish length. The oxygen concentration is calculated using the following equation:(10)$$Y_{cvss}=[O_{2}{]}_{cage}-\left(\dot{M}O_{2}U\;\cdot \rho \cdot v^{-1}\right),$$
where *Y_cvss_* is the oxygen concentration during a given current velocity and swimming speed, [O_2_]_cage_ is the oxygen concentration in the cage, $\dot{M}O_{2}$U is the oxygen consumption during swimming, *ρ* is the fish density, v^−1^ is the time for the current to move one meter. If swimming speed and current velocity are the same, $\dot{M}O_{2}$U is directly correlated with the velocity, and if fish are swimming at U_pref_, $\dot{M}O_{2}$U is unchanged until the current velocity equals or exceeds U_pref_.

During hypoxia, feed is sub-optimally utilized, and thresholds for the onset of the decrease are reported for rainbow trout at 15 °C to be at 6.0 mg O_2_ L^−1^ (Sal: 0, approx. 68.9% air sat. DO) [28]. Similar values were found for Atlantic salmon (*Salmo salar*) at 11, 15, 19 °C with limits at 53 ± 1, 66 ± 3 and 76 ± 4% air sat. DO, respectively [29]. The proximity of the two studies’ results suggests that air sat. DO thresholds are similar for these two species of salmonids. Fitting the values from both studies to a linear regression resulted in the following equation.
(11)$$Threshold\left(\%\;DO\right)=2.875\;\cdot T+22.85,$$
where T is the temperature.

### 2.5. Case Study Using the Model

The case study was based on a standard to large aquaculture sea cage during tidal change with low currents and at three different temperatures (10, 15, and 20 °C) with a salinity of 30 ppt and an atmospheric pressure of 1 atm. The mass-adjusted swimming energetic values used at 10, 15, and 20 °C were SMR of 54.8, 60.5, 99.8 mg O_2_ kg^−1^ h^−1^ with corresponding exponents at 1.114, 0.909, 0.646, respectively. The average body mass was 1.5 kg with a length of 45.0 cm. Using a circumference of 160 m and a depth of 8 m, the cage volume was 16,297.5 m^3^ and a diameter of 50.9 m. The fish density within the cage was 25 kg/m^3^, and SDA was estimated to be 40% of SMR. For the second calculation series in the model, the current velocity was 5 cm/s, and the swimming energetics were based on unfed and fed fish swimming at U_pref_.

### 2.6. Statistics

All statistical analyses were performed using SPSS version 27.

To check for possible carryover effects in SMR during swimming energetics in normoxia, an independent *t*-test was used on all temperatures. A test on means between temperatures was performed on the SMR, AMR, and Upref data from the normoxia experiments.

The means from the P_crit_ at 0.4, 1.0 BL·s^−1^ experiments and the minimum required pO_2_ for swimming at U_pref_ in the hypoxia experiments between temperatures were tested. 

The statistical level of significance was set at ⍺ = 0.05 for all statistical tests.

## 3. Results

### 3.1. Swimming Energetics—Normoxia

Results from the normoxia swimming experiments are illustrated in Figure 1A–C and summary of results reported in Table 1. SMR was determined by extrapolating the fitted swimming equation to a swimming speed of 0 BL·s^−1^. Since there was a considerable mass difference between the fish used in each temperature, a mass size correction was applied (Equation (5)). The body mass adjusted values are used in the model. The Q10 was 2.0 and 1.8 in the measured and adjusted SMR, respectively. The AMS was highest at 15 °C. The calculated U_opt_ at 15 °C was 1.1 BL·s^−1^ and this value is used as Upref for the model at all temperatures. 

### 3.2. Swimming Energetics—Hypoxia

A summary of results from hypoxia trials is shown in Figure 1D–F and Table 2. 

While the fish were swimming at 0.4 BL·s^−1^, the critical oxygen tension showed an increasing trend with temperature. This was also the case with fish swimming at 1.0 BL·s^−1^ from 10 to 20 °C. Using the Hill equation (Equation (4)), the minimum required oxygen tension for fish swimming at Upref was determined for each temperature. At 15 °C, the average minimum requirement is lower than the calculated requirements for 10 and 20 °C.

### 3.3. SDA Experiments

Experiments to quantify gas exchange at the surface of the tank ran for 8 h but showed no difference in oxygen tension throughout the period. This might be explained by the lack of vertical mixing and the oxygen being measured in the middle of the water column. Although the movement of fish in the tank will add some vertical mixing, it is unlikely to have changed the ambient oxygen tension in the water considerably and therefore gas exchange at the surface is excluded as a factor in the present study.

Regular feeding with 24 h intervals showed a clear equilibrium in $\dot{M}O_{2}$ being reached after 6 days and sustained until day 12. Feeding was halted on day 11, but $\dot{M}O_{2}$ remained unchanged for another 24 h, before decreasing back to RMR, which was reached on day 16 (Figure 2). The experiment was terminated at midday on day 17. 

### 3.4. Case Study Based on Model

Results from the case study are illustrated in Figure 3 and summarized in Table 3.

The model showed that in a cage with no current and fed fish swimming at U_pref_ (CS1), the oxygen content reaches the DO threshold three times faster with a temperature increase of 10 °C. With an increase of 10 °C, the oxygen concentration at the downstream edge of the cage with a current velocity of 5 cm/s and a fish density of 25 kg/m^3^ (CS2) decreased by 22.8%. The model also illustrated that if fish at the downstream edge of the cage are to utilize feed optimally at 20 °C, the cage diameter should not exceed 29.6 m. Finally, with a temperature increase of 10 °C, the minimum flow through the cage to sustain optimal feed utilization is increased by 4.1 cm/s (CS3).

### 3.5. Statistics

#### 3.5.1. Carryover Effects on SMR

SMR data in 10, 15 and 20 °C showed homoscedasticity (Levene’s, F = (0.01, 0.08, 0.93), *p* = (0.93, 0.79, 0.36)), respectively. There were no statistically significant differences between SMR means as determined by independent *t*-tests for 10, 15 and 20 °C were t(8) = (−1.34, 0.11, −0.30), *p* = (0.22, 0.92, 0.78), respectively. Hence, no carryover effect was present in SMR between experiments.

#### 3.5.2. Normoxia Experiments

Data for the SMR, AMR and optimal swimming speed (Uopt) at 10, 15 and 20 °C were normally distributed (Shapiro–Wilk: SMR: W(28) = 0.98, *p* = 0.88; AMR: W(28) = 0.99, *p* = 0.97; Uopt: W(28) = 0.96, *p* = 0.36) and showed equal variance (Levene’s test, SMR: F(2,25) = 1.77, *p* = 0.19; AMR: F(2,25) = 2.08, *p* = 0.15; Uopt: F(2,25) = 0.8, *p* = 0.46). There was a significant difference between the means of SMR, AMR and Uopt (one-way ANOVA, SMR: (F(2, 25) = 27.7, *p* < 0.01); AMR: (F(2,25) = 4.51, *p* = 0.02); Uopt: (F(2,25) = 21.8, *p* < 0.01)). The Tukey HSD test showed that statistically there was no difference in the means of SMR and Uopt at 10 and 15 °C. For AMR, the Tukey HSD test showed no significant difference between means at 15 and 20 °C.

#### 3.5.3. Hypoxia Experiments

In the hypoxia experiments, Pcrit at 0.4 and 1.0 BL·s^−1^ at 10, 15 and 20 °C were normally distributed (Shapiro–Wilk test, 0.4 BL·s^−1^: W(28) = 0.95, p = 0.17; 1.0 BL·s^−1^: W(28) = 0.96, *p* = 0.43), but only the 1.0 BL·s^−1^ showed equal variance (Levene’s test, 0.4 BL·s^−1^: F(2,25) = 4.78, *p* = 0.02; 1.0 BL·s^−1^: F(2,25) = 2.0, *p* = 0.16). Using the Brown–Forsythe test the 0.4 BL·s^−1^ (F = (2,14.21), *p* = 0.054) showed no significant difference between means. This was also the case with 1.0 BL·s^−1^ data (one-way ANOVA, F(2,25) = 2.7, *p* = 0.09).

The difference between the minimum required pO_2_ for swimming at Upref at different temperatures was not normally distributed (Shapiro–Wilk: W(28) = 0.92, *p* < 0.05), and therefore the nonparametric Kruskal–Wallis test was used (X2(2) = 8.62, *p* < 0.05) and showed there to be a significant difference between groups. A Mann–Whitney U test only indicated a statistical difference between the pO_2_ Limit for swimming at U_pref_ at 15 and 20 °C, U(N15 °C = 10, N20 °C = 8) = 20, z = −1.78, *p* < 0.01).

## 4. Discussion

### 4.1. Swimming Energetics—Normoxia

The standard metabolic rate increased with temperature from 10 to 20 °C, resulting in a Q10 of 2.0 and 1.8 in the absolute and body mass adjusted values, respectively. Although a lower Q10 was found for adjusted SMR, it is still within the range for fishes 1.62 to 2.64 [8,19,20,21,22]. Significant increases were seen in SMR between 15 to 20 °C and between 10 to 15 °C in AMR. This reflects the AMS, which was greatest at 15 °C and is in accordance with T_opt_ for rainbow trout, which at normoxia is approx. 15.8–17 °C [44,45,46]. At T_opt_, the aerobic scope (AMS) is highest, i.e., the fish will have the highest surplus energy and optimal energy efficiency.

### 4.2. Swimming Energetics—Hypoxia

The mean critical oxygen tension for swimming at 0.4 and 1.0 BL·s^−1^ was not significantly different between temperatures. The same pattern was not seen at 1.0 BL·s^−1^, where the critical oxygen tension was unchanged or even decreased from 10 °C to 15 °C. This might be explained by the increase in AMS in proximity to T_opt_, resulting in increased hypoxia resilience.

### 4.3. SDA Experiments

The duration and magnitude of SDA is affected by a multitude of both internal and external factors [23,24], among which, temperature stands as a pivotal factor, as elevated temperatures can increase metabolic processes, resulting in accelerated digestion. This phenomenon persists until the temperature reaches a threshold, leading to stress-induced conditions, which can prolong the SDA duration. Additionally, fish sizes, feed composition, size, quantity, and feeding frequency represent other influential determinants of SDA. Therefore, it is important to acknowledge the inherent bias associated with utilizing the percentage of SMR to delineate SDA equilibrium in the model.

A more advanced approach to SDA could have resulted in more precises model predictions but due to the model being based on cold water species, the overall change would have been minimal. The authors therefore have chosen a simple approach, just using a percentage of SMR as an indication of the increase in $\dot{M}O_{2}$ caused by SDA. Furthermore, the SDA experiment in the present study showed that if the interval between feedings is less than the digestion time, $\dot{M}O_{2}$ caused by SDA will reach an equilibrium. It is likely that this equilibrium will change based on the above-mentioned factors, but they would be difficult to quantify for any given scenario, hence they are not included in the present study.

### 4.4. The Model and Case Study

Models are valuable tools for researchers to explain aspects of nature that are difficult to grasp, and the field of aquaculture research is no different. Models for growth performance [47] and behavior [48,49] have been created for Atlantic salmon in sea cages. Furthermore, models similar to the model in the present study, that predict oxygen levels within sea cages, based on fish density, activity and cage sizes have also been created [50,51]. The distribution of oxygen concentrations within sea cages has been reported to fluctuate vertically [52] and being affected by cage sizes [53] and fish densities [54]. Our model was created to assess the limits for fish density, minimum flow velocities and cage sizes based on environmental and biological parameters. The case study using the model showed that with increasing temperatures and fish preferring to swim at U_pref_, the likelihood of the oxygen concentration reaching sub-optimal conditions is high. If periods of hypoxic conditions occur within the cage, the lower oxygen content will have a limiting effect on aerobic swimming since AMR and thereby the AMS are reduced [55]. Oxygen concentration below the DO threshold [28,29] for feed utilization will also cause digestion to be sub-optimal, and if the decrease continues, appetite will be reduced [30,31]. This will cause a financial loss for the fish farmer and increase the harmful effects of farming on the surrounding environment, due to feed sinking out of the cage.

Many of the challenges proposed in the present study can be overcome if the current velocity is maintained above the flow threshold found in CS3 or if durations of low water flow are short (CS1). These challenges are especially apparent with high temperatures.

### 4.5. Limitations and Model Uncertainties

Modelling advanced systems, such as nature, will always have elements of simplification in them. These simplifications cause limitations and uncertainties in the model predictions. In the present study, the major simplifications were SDA, cage proximities, fish distribution, and effects on fish behavior/activity caused by operations on the farms. As for SDA, working with cold water species, the increase in $\dot{M}O_{2}$ constitutes a relatively small part of the AMS; hence the difference in the model prediction would be minimal. Therefore, a simple approach was chosen, which anyone can follow.

Since the model only focuses on a single sea cage, factors such as low-oxygen water from neighboring cages entering the model cage are not included. If the proximity between cages is small, mixing between cages might not be sufficient and hence the oxygen concentration of inflow water will be lower than modelled. This is also a limitation of the model.

In the present study, fish density was based on the size of the cage, hence given the fish density a uniform distribution throughout the cage. This is not a likely scenario on a fish farm, since normal operations on the farm, such as feeding, and maintenance will increase the activity of the fish. Fish distribution within sea cages has been shown to vary based on environmental factors [54] and therefore the chosen uniform distribution can only show a best-case scenario, hence we can only assume that conditions are worse than the model predicts. Furthermore, the increased activity of the fish during feeding will further increase $\dot{M}O_{2}$ causing an even greater oxygen demand and if water exchange is limited, a decrease in oxygen concentrations within the cage is inevitable.

### 4.6. Applications of the Model

There are a multitude of possible applications for the model, ranging from aquaculture industry optimization to teaching purposes. For the aquaculture industry, the model can be used on both existing farms and potential new farming locations. At farm sites, measurements of oxygen may vary greatly depending on where they are measured. It is not uncommon that oxygen is measured next to the feeding-barge. This approach can only inform the farmer of the oxygen concentration outside of the cages and not the oxygen availability within the cage. By combining chemical and physical measurements for the site as well as fish density and species energetics, the model can give a prediction of the oxygen availability within the cage. This way, the farmer will be informed if oxygen decreases below the DO threshold for optimal feed utilization. Furthermore, using the measured chemical and physical parameters with swimming energetics the model can inform the farmer of the maximum cage sizes for a given fish density and vice versa. Using the same method, the model can also estimate if a potential new farming site is viable. Although average measurements of chemical and physical parameters are important to estimate how well a potential area will work for farming, extremes are equally important, since they might be the source for high mortality in the farm. In contrast to fish living in their natural habitat, farmed fish do not have the option to seek refuge from stressors, exemplified by substantial temperature fluctuations and excessive water current velocities.

One option that has been proposed in recent years is moving aquaculture offshore to more exposed areas [56]. This would solve most potential issues with oxygen availability since water exchange is increased in the cages but might also expose fish to excessive water current velocities. Other benefits are the dispersal of discharge from the farms, which will be mixed into the open water masses and hence reduce effects on the surrounding environment. Although mostly beneficial for the farmer, the increased distance from land and the harsher weather might prove to be challenging.

Finally, understanding the interaction between the variables involved in oxygen availability in an aquaculture sea cage might be difficult for students. The model gives an opportunity to change the variables and see how the oxygen is affected within the cage. To further increase the availability of the model, a graphical user interface is available on the project webpage (https://www1.bio.ku.dk/english/research/mbs/projects/fitfish/, last accessed: 30 September 2023, DOI: 10.5281/zenodo.4629693).

## Figures and Tables

**Figure 1 biology-12-01408-f001:**
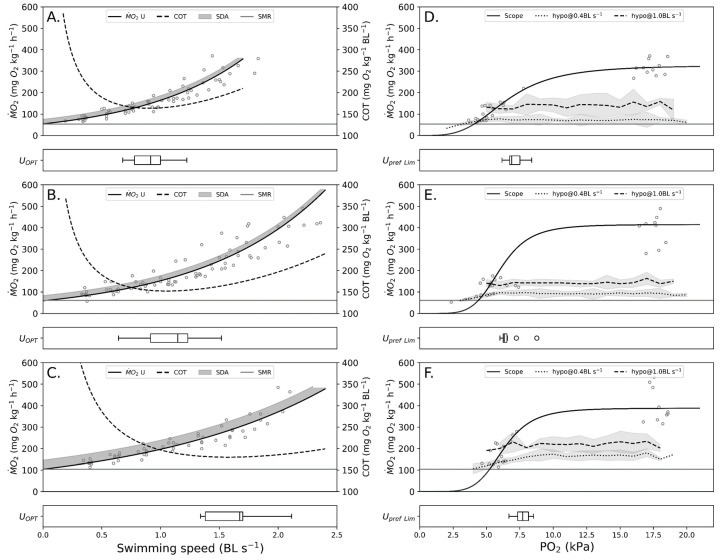
Swimming energetics in normoxia and hypoxia. Plots (**A**–**C**) illustrate swimming energetics (-) with SDA (shaded area) from the normoxia experiments at 10, 15, and 20 °C, respectively. Cost of transport (COT) is also illustrated on each plot with corresponding horizontal bars showing the average optimal swimming speed. Plots (**D**–**F**) show $\dot{M}O_{2}$ measurements from the hypoxia experiments, at 10, 15, and 20 °C, respectively. Swimming energetics in hypoxia at 0.4 and 1.0 BL s^−1^ and AMR are fitted to the Hill equation. Finally, the Hill equation is used to estimate the pO_2_ limits for Upref swimming. These are illustrated as horizontal bars with standard deviation.

**Figure 2 biology-12-01408-f002:**
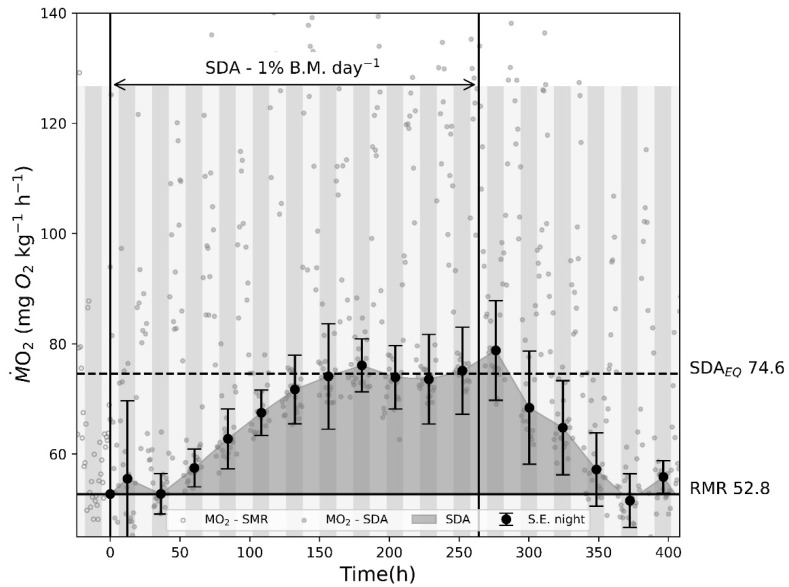
SDA experiment while fish are fed 1% of TBM each day until feeding is halted. A clear equilibrium is present after the 6th day of feeding. After feeding is halted, $\dot{M}O_{2}$ caused by SDA stay’s elevated for another 24 h, before returning to RMR. Average $\dot{M}O_{2}$ measurements for RMR and SDA at equilibrium were 52.8 and 74.6 mg O_2_ kg^−1^ h^−1^, respectively, which constitutes an increase of 41.3%. The authors have therefore chosen a percentage of 40% of SMR as the benchmark increase in oxygen consumption caused by SDA in the model independent of temperature and fish size.

**Figure 3 biology-12-01408-f003:**
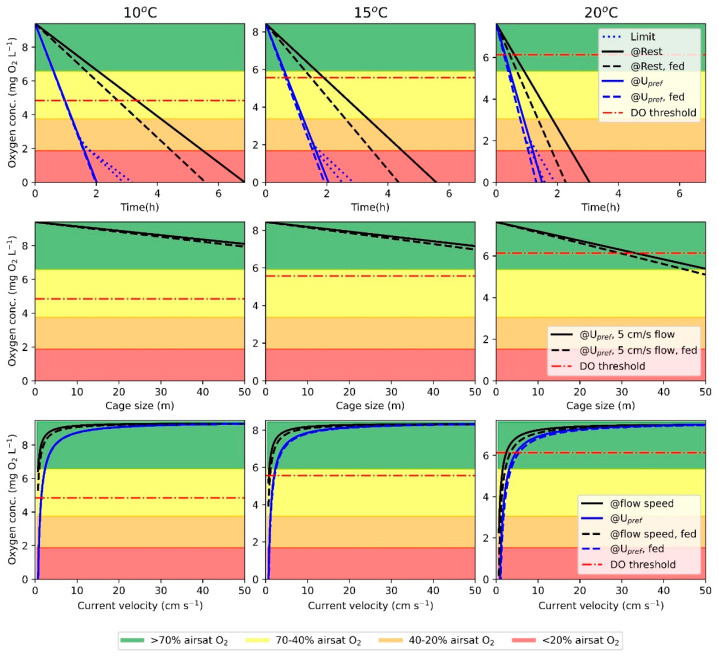
Results from the case study using the model. The three columns show temperature (10, 15, and 20 °C). On each plot, the DO threshold for suboptimal feed utilization is illustrated as horizontal lines (Dot-dashed). The first row is the duration to oxygen depletion (CS1) in fed and unfed fish at rest or swimming at U_pref_. When oxygen concentration decreases below the minimum requirement for swimming at U_pref_, the fish will decrease swimming speed and thereby oxygen consumption. The second row of plots illustrates oxygen concentration through the cage in fed and unfed fish, swimming at U_pref_ with a current velocity of 5 cm/s (CS2). The final row of plots illustrates oxygen concentration when fed and unfed fish are swimming at the same speed as the current velocity and when fish are swimming at Upref until swimming speed matches current velocity, then swimming speed and current velocity are the same (CS3).

**Table 1 biology-12-01408-t001:** Summary of results from swimming energetics for rainbow trout in normoxia (Mean ± SD).

	10 °C	15 °C	20 °C
Fish (n)	10	10	8
Mass (g)	1997.0 ± 338.7	1373.3 ± 333.4	1075.0 ± 124.0
Total length (cm)	50.6 ± 1.6	46.5 ± 2.4	44.6 ± 1.6
SMR (mg O_2_ kg^−1^ h^−1^)	53.1 ± 10.7	60.5 ± 20.0	104.6 ± 10.4
SMR_1.5kg-adj._	54.8 ± 12.3	60.1 ± 21.8	100.7 ± 10.4
AMR (mg O_2_ kg^−1^ h^−1^)	298.2 ± 50.2	414.9 ± 100.5	393.1 ± 102.3
FAMS	5.9 ± 2.0	7.8 ± 5.0	3.8 ± 1.0
U_crit_ (BL·s^−1^)	1.6 ± 0.2	2.1 ± 0.3	2.0 ± 0.3
U_opt_ (BL·s^−1^)	0.90 ± 0.17	1.10 ± 0.27	1.62 ± 0.26

**Table 2 biology-12-01408-t002:** Results from swimming energetics in hypoxia (mean ± SD). All P_crit_ values are partial pressure of oxygen (pO_2_).

	10 °C	15 °C	20 °C
P_crit_@0.4 BL·s^−1^	4.49 ± 0.56	5.12 ± 1.31	5.48 ± 1.27
P_crit_@1.0 BL·s^−1^	5.79 ± 0.97	5.43 ± 0.57	6.33 ± 0.67
P_crit_ @U_pref (min req)_	7.08 ± 0.67	6.64 ± 0.78	7.70 ± 0.58

**Table 3 biology-12-01408-t003:** Summary of results from the case study using the model. CS refers to the Calculation Series discussed in Section 2.4.4. and DO threshold is suboptimal feed utilization. CS1 is oxygen decrease over time with no current, CS2 is oxygen decrease through the cage with flow velocity at 5 cm/s, and CS3 is oxygen concentration affected by flow and swimming simultaneously. All results are based on fed fish swimming at Upref.

	10 °C	15 °C	20 °C
Oxygen solubility (mg O_2_ L^−1^)	9.39	8.44	7.63
DO threshold (mg O_2_ L^−1^)	4.86	5.57	6.13
CS1: DO threshold (min)	58	36	19
CS2: Downstream edge (mg O_2_ L^−1^)	7.9	6.9	6.1
CS2: DO threshold _max cage_ (m)	155.7	98.0	29.6
CS3: DO threshold (flow: cm s^−1^)	1.5	2.2	5.6

## Data Availability

The result of the paper, in form of the model is available at 10.5281/zenodo.4629693. All other results are available in the paper as clear text.

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
