# Peer review of "Model of Oxygen Conditions within Aquaculture Sea Cages"

_biology, 2023, doi:10.3390/biology12111408_

Round 1

Reviewer 1 Report

Comments and Suggestions for Authors

General comments:

The manuscript presents an interesting study ranging from 1) laboratory experiments to assess the effects of temperature, current and swimming speed and metabolic activity on oxygen consumption rate (MO2) to 2) simple models describing oxygen conditions in cages as functions of those variables. I believe the manuscript should be published, but revision is needed, as indicated in my comments below.

Introduction:

The authors give a good overview of the factors that influence MO2. In my opinion there is room for improvement in the description of how MO2 interacts with other factors to determine oxygen availability in sea cages.

I am skeptical of the following statement: "As ṀO2 increases exponentially with increasing swimming speed [10], even slight increments in swimming speed during low current velocities will profoundly affect oxygen availability within the cage.": Such a claim needs to be better founded. To make it less vague, you might define more precisely what you mean by "slight" and "profound".

In the description of what factors influence oxygen availability, I was surprised to find no references to this recent oxygen model that is highly relevant to the authors' study:
- Alver, M.O., Føre, M. and Alfredsen, J.A., 2022. Predicting oxygen levels in Atlantic salmon (Salmo salar) sea cages. Aquaculture, 548, p.737720.
- Alver, M.O., Føre, M. and Alfredsen, J.A., 2023. Effect of cage size on oxygen levels in Atlantic salmon sea cages: A model study. Aquaculture, 562, p.738831.

Materials and methods:

Experiments run to determine swimming energetics, critical swimming speed, including the effect of hypoxia, and the effect of SDA on MO2 are clearly presented in sufficient detail.

The model: 

Cage volume is used to calculate fish density, under the assumption of uniform distribution. However, hypoxia is more likely to occur in cases where the fish distribution is skewed towards parts of the cage volume, so what is the rationale behind this choice?

Eq. (10): What is the last factor on the right side? Is this a "v" or another symbol? If it is a "v" for current velocity, does it not cancel against the previous factor (inverse of velocity)? If the symbols are supposed to be different, I'd recommend using more distinctive symbols. If they are both the symbol for velocity, please type them in the same way to avoid confusion.

I don't understand what Eq. (10) describes. Higher fish density will apparently give higher oxygen, as will higher MO2? Am I missing something here? Please describe how you arrived at this equation.

I am also confused as to why Eq. (10) apparently describes a static situation, while Eq. (8-9) describes time-variable and spatially variable values. How do these models relate to each other?

Results:

Fig. 1: the figure seems to have low resolution, some details are hard to make out. This is also the case for the other figure.

3.1: As noted by the authors, there is a large difference in the average mass of fish used at the three temperatures. Why were larger fish used for low temperatures and smaller fish for high temperatures? A discussion of potential confounding factors related to this would be useful.

3.3: The gas exchange experiment, and its results, are interesting. It would be interesting to se an experiment with artificially induced turbulence in the tank, to test if oxygen exchange would significantly increase (this is just a side comment, I'm not requesting this be added to the present study).

Discussion:

4.4: "Models for growth performance [47] and behavior [48], [49] have been created for Atlantic salmon in sea cages, but to the author’s knowledge, no models have been developed that predict oxygen availability within sea cages.": At least one such model exists, see earlier comment. I would welcome discussion of how the results of the case study compare with the results of Alver et al. (2023).

Regarding model limitations, the assumption of uniform fish biomass should be mentioned, as well as the current speed being treated as a uniform value - both of which may not always be good assumptions. In particular, feeding may strongly affect the distribution of fish, and thus potentially representing the most critical conditions with regard to oxygen availability. Another limitation that is not mentioned is the presence of neighboring cages, which may significantly affect the oxygen levels of the water entering a cage.

The case study is interesting, but hard to evaluate since no comparisons to actual observations are made, and the discussion of the model uncertainty is very brief. To address uncertainty, a discussion of the model's sensitivity to parameters and inputs should be included.

Reviewer 2 Report

Comments and Suggestions for Authors

Decision on the manuscript with ID (biology-2680617) is “Major Revisions”. The authors should prepare a point-by-point response to the comments raised by the anonymous reviewer. The authors should follow the comments presented in the PDF file.

Q1. How long did fish take to be acclimated to salinity? What is the salinity gradient per day?

Q2. How fish acclimation took place? Add details.

Q3. What are steps and procedures performed to monitor the general health status of the acclimated fish?

Q4. Did you perform bacteriological, fungal, or parasitological examinations? Add all the details and also the results.

Q5. Add details about water exchange procedures and exchange rate.

Minor comments: -

Q1. In Abstract: Add the values of DO (mg/L) for cases of normoxia and hypoxia.

Q2. Follow the journal guidelines for writing citations in the text. Tudorache et al. [11] and Eliason and Farrell [17].

Q3. Reference section: Revise Journal names. Latin names should be written italic (see the PDF file).

Comments on the Quality of English Language

Moderate editing of English language required

Reviewer 3 Report

Comments and Suggestions for Authors

Reviewer Comments:

I had the privilege of reviewing your manuscript titled "Model of oxygen conditions within aquaculture sea cages", which presents research on swimming energetics, oxygen availability in aquaculture sea cages, and the development of a predictive model. I commend your efforts in conducting this research, and I appreciate the contributions your work can make to aquaculture. The English language usage in the manuscript is generally quite good. The text is clear and mostly well-structured, making it easy to follow the research and the development of the model. I want to provide detailed feedback and comments, both appreciation and critical points, to assist you in enhancing the quality and clarity of your manuscript. The manuscript is overall well-structured, and the use of section headings and figures significantly aids in comprehending the research design, data, and outcomes. Your study encompasses comprehensive experiments on swimming energetics in both normoxia and hypoxia, as well as Specific Dynamic Action (SDA) experiments. These experiments provide a strong foundation for developing your subsequent model and case study. The model you have developed to predict oxygen availability within sea cages is a significant contribution to aquaculture research. Its practical applications for both existing farms and potential new farming locations, offering insights into fish behavior and oxygen conditions, are commendable. Your study explores the impact of temperature on metabolic rates, critical oxygen tensions, and fish behavior. The findings emphasize the importance of considering temperature effects in aquaculture management and expand our understanding of these dynamics. The inclusion of a graphical user interface for your model enhances its accessibility and usability. This addition makes your research tool not only valuable for researchers but also for students and individuals interested in aquaculture.

I would be happy if you could answer the following queries and/or incorporate some of the suggestions  (Minor revision to the manuscript) here:

 1.      While your study discusses the Q10 values and the impact of temperature on metabolic rates, a more detailed discussion of the ecological and physiological implications of these findings could provide a richer context. How do these temperature-related changes affect fish in their natural environments and in aquaculture settings?

2.      You employ a simplified approach to modeling Specific Dynamic Action (SDA) by assuming a fixed percentage increase in metabolic rate. While this may be adequate for the study's purposes, a discussion of the limitations and potential sources of variability in SDA would be beneficial.

3.      Your study touches on fish behavior but does not delve deeply into the behavioral responses of fish to changing oxygen conditions. A more thorough exploration of the behavioral aspects and their potential impacts on aquaculture operations could provide valuable insights. If there are plans further on the same (if another manuscript is being made ready in line with this), it can be ignored in this manuscript.

4.      You mention that external factors can affect SDA and fish behavior but do not elaborate on what these factors are. Providing examples or discussing potential external factors would enhance the study's comprehensiveness.

5.      While the limitations of the model are acknowledged, a more explicit discussion of the specific uncertainties associated with the model's predictions would be useful. What are the potential sources of error, and how might these affect the model's accuracy? This may help other researchers working similarly in this thematic area.

6.      Your study involves multiple aspects, including biology, modeling, and aquaculture. A more integrated discussion of how these fields interact and contribute to the overall understanding of oxygen availability in sea cages would be beneficial.

I want to emphasize that your research is indeed valuable and has the potential to make a significant impact in the field of aquaculture. Addressing the above points (Minor revision) will enhance the clarity and depth of your manuscript, making it an even more valuable resource for researchers, practitioners, and students.

 Regards,

The other reviewer.

Round 2

Reviewer 2 Report

Comments and Suggestions for Authors

The authors properly addressed the comments raised by the reviewer.

Comments on the Quality of English Language

Minor editing of English language required